# The acquisition of molecular drivers in pediatric therapy-related myeloid neoplasms

Jason R. Schwartz[1,10], Jing Ma[2,10], Jennifer Kamens[3,10], Tamara Westover[2], Michael P. Walsh[2], Samuel W. Brady [4], J. Robert Michael [4], Xiaolong Chen [4], Lindsey Montefiori[2], Guangchun Song[2], Gang Wu [4], Huiyun Wu[5], Cristyn Branstetter[6], Ryan Hiltenbrand[2], Michael F. Walsh [7], Kim E. Nichols [8], Jamie L. Maciaszek[8], Yanling Liu[4], Priyadarshini Kumar[2], John Easton[4], Scott Newman[4], Jeffrey E. Rubnitz[8], Charles G. Mullighan [2], Stanley Pounds [5], Jinghui Zhang [4], Tanja Gruber[3,9✉], Xiaotu Ma [4✉] & Jeffery M. Klco [2✉]

Pediatric therapy-related myeloid neoplasms (tMN) occur in children after exposure to cytotoxic therapy and have a dismal prognosis. The somatic and germline genomic alterations that drive these myeloid neoplasms in children and how they arise have yet to be comprehensively described. We use whole exome, whole genome, and/or RNA sequencing to characterize the genomic profile of 84 pediatric tMN cases (tMDS: *n* = 28, tAML: *n* = 56). Our data show that Ras/MAPK pathway mutations, alterations in *RUNX1* or *TP53*, and *KMT2A* rearrangements are frequent somatic drivers, and we identify cases with aberrant *MECOM* expression secondary to enhancer hijacking. Unlike adults with tMN, we find no evidence of pre-existing minor tMN clones (including those with *TP53* mutations), but rather the majority of cases are unrelated clones arising as a consequence of cytotoxic therapy. These studies also uncover rare cases of lineage switch disease rather than true secondary neoplasms.

[1] Vanderbilt University Medical Center, Department of Pediatrics, Nashville, TN, US. [2] St. Jude Children's Research Hospital, Department of Pathology, Memphis, TN, US. [3] Stanford University School of Medicine, Department of Pediatrics, Stanford, CA, US. [4] St. Jude Children's Research Hospital, Department of Computational Biology, Memphis, TN, US. [5] St. Jude Children's Research Hospital, Department of Biostatistics, Memphis, TN, US. [6] Arkansas Children's Northwest Hospital, Department of Hematology/Oncology, Springdale, AR, US. [7] Memorial Sloan Kettering Cancer Center, Department of Pediatrics, New York, NY, US. [8] St. Jude Children's Research Hospital, Department of Oncology, Memphis, TN, US. [9] Stanford University School of Medicine, Stanford Cancer Institute, Stanford, CA, US. [10] These authors contributed equally: Jason R. Schwartz, Jing Ma, Jennifer Kamens. ✉email: tagruber@stanford.edu; xiaotu.Ma@stjude.org; jeffery.klco@stjude.org

Although the therapeutic regimens for pediatric cancer have improved with a resultant overall decrease in the incidence of tMN in children[1–4], approximately 0.5–1.0% of children continue to develop tMN after therapy for hematological, solid, and CNS malignancies[2]. Children with tMN have a worse prognosis compared to de novo MDS/AML, with 5-year survival rates of 6–11% if not treated with hematopoietic cell transplant (HCT)[1,2]. While much effort has focused on tMN in adults[5–9], a complete understanding of the pathogenesis of tMN in children is lacking despite well-described associations with alkylating agents (e.g., cyclophosphamide), topoisomerase II inhibitors (e.g., the epipodophyllotoxins etoposide and teniposide), radiation therapy, and HCT[10–14]. Epipodophyllotoxin-associated tMN is strongly associated with KMT2Ar[10,15].

Here, using a comprehensive sequencing approach, we show that Ras/MAPK pathway mutations, alterations in RUNX1 or TP53, and KMT2A rearrangements are frequent somatic drivers in pediatric tMN, and we find that in some cases aberrant MECOM expression is secondary to enhancer hijacking. Additionally, using samples from serial timepoints, we find no evidence of pre-existing minor tMN clones (including those with TP53 mutations) like in adults with tMN[5–7], but rather the majority of cases are unrelated clones arising as a consequence of cytotoxic therapy.

## Results

**Sequencing of pediatric tMN samples.** Eighty-four pediatric tMN cases, including tMDS ($n = 28$) and tAML ($n = 56$), were profiled, including both tumor and non-tumor tissue for 62 cases and only non-tumor material for 22 cases (Table 1 & Supplementary Data 1). Initial diagnoses included hematologic (70%), solid (27%), and brain (3%) neoplasms (Fig. 1a). The median age at tMN was 13.6 years (range: 1.2–24.6 yrs) (Supplementary Fig. 1a, b, & Supplementary Data 2), and the time to tMN after initial diagnosis varied widely (median: 2.9 yrs; range: 0.7–16.2 yrs) (Supplementary Fig. 1c–e, & Supplementary Data 3). Somatic variants identified from WGS (median coverage: 50x) or WES (112x) were validated by targeted resequencing (641x) (Supplementary Data 4–8).

A mean of 28 (range: 1–188) somatic mutations per patient were identified, which is significantly greater than the mutational burden found in pediatric primary MDS (5 mutations/patient, $p < 0.001$) and pediatric de novo core-binding factor AML (13 mutations/patient, $p < 0.001$)(Fig. 1b)[16,17]. Four patients had mutation burdens greater than 2 standard deviations above the mean, ranging from 115 to 188 mutations/patient (Supplementary Fig. 2a). We detected DNA repair pathway gene (PMS2; $n = 2$, MSH6; $n = 1$) alterations in 3 of these hypermutated cases (Supplementary Data 9). In the fourth case (SJ016473), the hypermutation status appears to be driven by variants with variant allele frequency (VAF) < 0.2 (Supplementary Fig. 2b), and the corresponding driver alteration could have escaped detection due to limited depth. Including multiple modes of somatic

alterations (SNV, CNV, & fusions), we used the Genomic Random Interval (GRIN) model[18] to identify 91 genes that were significantly altered in this cohort (Supplementary Data 10). The most common altered functional pathways were epigenomic ($n = 57$ of 62, 92%) and cell signaling ($n = 46$ of 62, 74%), with mutations in the Ras/MAPK pathway, including KRAS and NF1, and mutations or structural alterations involving RUNX1 and KMT2A being the most frequent (Fig. 1c,d, & Supplementary Data 11).

**Putative germline variants in pediatric tMN.** Fourteen pathogenic or likely pathogenic presumed germline sequence alterations were identified in 13 of 84 patients (15%, 95% exact binomial CI: 8.5–25.0%) (Table 2 & Supplementary Data 12–14), indicating that germline alterations may be more common in tMN than the published prevalence of 8.5–10% in other groups of children with cancer[19–22]. This includes 4 patients with germline TP53 mutations. There was also evidence of TP53 mosaicism in the non-tumor tissue in 5 additional patients (Fig. 1e & Supplementary Data 15). Collectively, 15 patients (18%) had somatic (mutation and/or copy number alteration) or germline alterations in TP53 (Supplementary Fig. 3). There was a significant enrichment of complex cytogenetics in patients with TP53 alterations (11 of 13) versus wild-type TP53 patients when considering those with comprehensive sequencing ($n = 62$, 85% vs. 12%; Fisher's $p < 0.0001$) (Supplementary Fig. 3e). Three other patients had low VAF somatic truncating mutations in exon 6 of PPM1D (Supplementary Fig. 4)[23,24]. Despite the fact that deletions or CN-LOH involving chromosome 7 (del(7)) were the most common copy number alteration (22 of 62, 35%) (Fig. 1f, Supplementary Fig. 5, & Supplementary Data 16), germline mutations in SAMD9, SAMD9L, GATA2, or RUNX1 were not present[16,25–27]. The comprehensive mutational profile of pediatric tMN is shown in Fig. 2a.

**Mutational signatures of pediatric tMN.** C > T transitions were the predominant mutation type (Fig. 2b, c). Mutational signature analysis on the 16 WGS cases and 3 WES cases with a sufficient quantity of SNVs (>30) identified drug signatures in 9 cases, including 4 with the cisplatin signature (COSMIC 31 & 35), and 5 with the thiopurine signature[28], consistent with the prior treatment history (Supplementary Data 17). Eight cases did not have a detectable drug signature but rather clock-like signatures 1, 5, and 40 (Fig. 2d)[29,30], while 2 additional patients had a signature similar to one of unknown etiology recently reported in relapsed mismatch repair (MMR)-deficient ALL[31] which we term the "relapse MMR" signature. Both had germline (SJ016519) or somatic (SJ016494) pathogenic PMS2 mutations. The relapse MMR signature bore similarities to the thiopurine signature (Supplementary Fig. 6), had similar strand bias to the thiopurine signature[28] (Supplementary Fig. 7), and occurred in patients with previous thiopurine exposure, thus suggesting it was a variant of the thiopurine signature that occurs under MMR-deficient conditions. We determined the probability that driver SNVs were caused by each signature as reported previously[28] (Fig. 2d, bottom), and found that 2 TP53 mutations were most likely (>50% probability) induced by cisplatin or thiopurines along with several Ras pathway and other variants. Example calculations showing the probability that specific driver mutations were caused by individual signatures are shown in Supplementary Fig. 8. These calculations are based on the signatures present in each sample and their mutation preference at specific trinucleotide contexts; thus, two KRAS G12D mutations in two different patients (SJ030799 and SJ016494) were likely caused by different

---

**Table 1 Sequencing Approach for the Pediatric tMN Cohort.**

|  |  | Cases | WGS | WES | RNA Seq |
|---|---|---|---|---|---|
| Unique patients |  | 84 |  |  |  |
| Tumor-normal pairs |  |  |  |  |  |
|  | tMDS | 23 | 3 | 23 | 19 |
|  | tAML | 39 | 13 | 35 | 37 |
| Normal only |  |  |  |  |  |
|  | tMDS | 5 |  | 5 |  |
|  | tAML | 17 |  | 17 |  |
|  | Total | 84 | 16 | 80 | 56 |

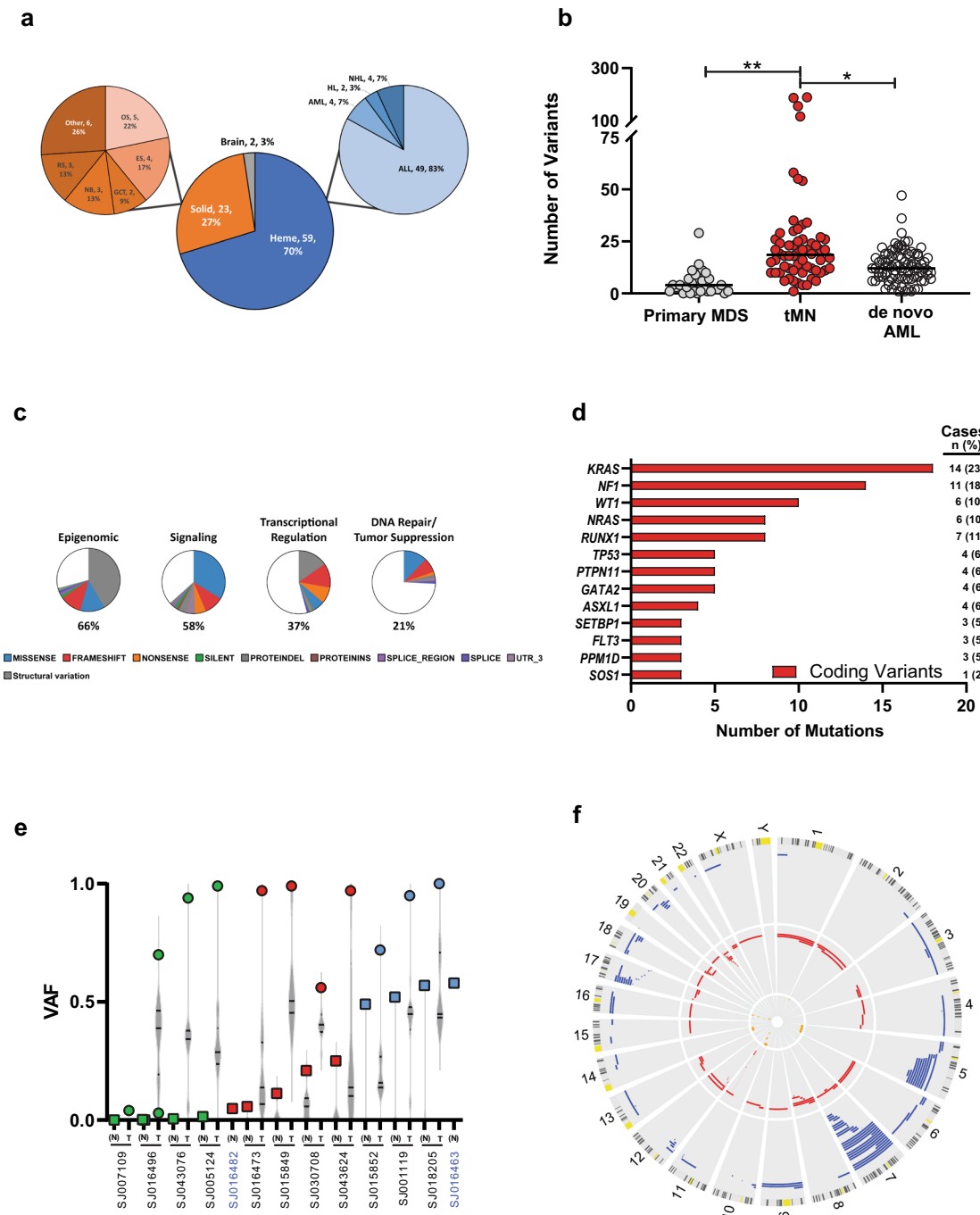

**Fig. 1 Clinical and genomic features of the pediatric tMN cohort. a** Pie charts depicting the distribution of initial diagnoses within the pediatric tMN cohort. AML acute myeloid leukemia, HL Hodgkin lymphoma, NHL non-Hodgkin lymphoma, ALL acute lymphoblastic leukemia, OS osteosarcoma, ES Ewing sarcoma, GCT germ cell tumor, NB neuroblastoma, RS rhabdomyosarcoma, Other includes: embryonal sarcoma, Wilms tumor, rhabdoid tumor, ovarian carcinoma, and peripheral neuroepithelioma. **b** Total number of somatic mutations per patient (includes the following mutation types: silent, nonsense, frameshift, indel, splice site, ITD, RNA coding genes, 3′ and 5′ UTR) compared to pediatric primary MDS[16] and de novo AML[17].*$p < 0.001$; **$p < 0.0001$. Black bar indicates the median. Wilcoxon–Mann–Whitney non-parametric, two-tailed test used to compare biologically independent samples from $n = 62$ tMN, $n = 32$ primary MDS, and $n = 87$ de novo AML cases. **c** Pie charts showing the distribution of recurrently mutated pathways in the pediatric tMN cohort and the distribution of mutation types within each pathway. Percentages refer to the frequency of mutations within a pathway amongst all somatic mutations present in the cohort. **d** The genes most frequently mutated (somatic) in pediatric tMN—Only coding variants are shown. **e** VAF plot showing the 13 patients with *TP53* mutations (SNV or indel). Tumor (T; circles) and normal (N; squares) are shown for each unique patient. Green symbols denote cases with VAFs suggesting somatic variants, blue symbols denote cases with clear germline variants in the normal tissue, and red symbols denote cases with *TP53* mosaicism. *$p < 0.01$ for binomial mosaicism test. Violin plots represent the range of VAFs for all somatic variants in that case. Black bars indicate the median and upper and lower quartiles. Note: SJ016482 and SJ016463 are from the normal only group of patients (blue font). **f** Circos plot showing copy number alterations found via WES ($n = 58$) & WGS ($n = 4$) analysis of 62 tumor/normal pairs. Circumferential numbers indicate chromosome number, blue lines = deletions, red lines = amplifications, and orange lines = CN-LOH.

**Table 2 Pathogenic and Likely Pathogenic Germline Variants Present in the Pediatric tMN Cohort.**

| Case | 1° Diagnosis | 2° Dx | Gene | RefSeq accession | Mutation type | Amino acid change | VAF | REVEL score | ACMG classification (criteria) |
|---|---|---|---|---|---|---|---|---|---|
| SJ016504 | NHL | tAML | ARID2 | NM_152641 | nonsense | p.R1272X | 0.53 | | LP (PVS1, PM2) |
| SJ016509 | ALL | tMDS | CREBBP | NM_004380 | missense | p.R1446C | 0.35 | 0.952 | LP (PS2, PM2, PP3) |
| SJ043618 | ALL | tAML | ETV6 | NM_001987 | nonsense | p.R359X | 0.56 | | P (PVS1, PS3, PM2, PP1) |
| SJ021960 | ALL | tMDS | ETV6 | NM_001987 | frameshift | p.N386fs | 0.30 | | P (PVS1, PS3, PM2) |
| SJ004031 | ALL | tMDS | EZH2 | NM_001203247 | missense | p.R685H | 0.43 | 0.907 | LP (PM2, PP2, PP3) |
| SJ016496 | ALL | tAML | NF1 | NM_000267 | nonsense | p.R2496X | 0.50 | | P (PVS1, PM2, PP1) |
| SJ016519 | ALL | tAML | PMS2 | NM_000535 | missense | p.S46I | 0.34 | 0.939 | LP (PS3, PP1, PM3, PP3) |
| SJ004031 | ALL | tMDS | PTPN11 | NM_002834 | missense | p.S502L | 0.39 | 0.976 | LP (PM1, PM2, PP2, PP3) |
| SJ043615 | ALL | tAML | RPL22 | NM_000983 | splice | E4Q_E3splice | 0.44 | | LP (PVS1, PM2) |
| SJ016463 | Osteosarcoma | tMDS | TP53 | NM_000546 | missense | p.R337C | 0.58 | 0.715 | P (PS3, PM1, PM2, PP2, PP3) |
| SJ001119 | Osteosarcoma | tAML | TP53 | NM_000546 | missense | p.R337L | 0.58 | 0.765 | P (PS3, PM1, PM2, PM5, PP3) |
| SJ015852 | ALL | tMDS | TP53 | NM_000546 | nonsense | p.W53X | 0.52 | | P (PVS1, PM2, PM2, PP4) |
| SJ018205 | Anaplastic Astrocytoma | tMDS | TP53 | NM_000546 | missense | p.H179Y | 0.50 | 0.948 | P (PS2, PS3, PM1, PM2, PP1, PP3) |
| SJ016486 | ALL | tAML | TRIP11 | NM_004239 | frameshift | p.Q1367fs | 0.40 | | LP (PVS1, PM2) |

mutational processes due to the presence of different signatures in the two samples.

**Chromosomal rearrangements present in pediatric tMN.** Chromosomal rearrangements encoding fusion oncoproteins were identified by RNA-seq in 70% of cases (39 of 56 with available RNA). *KMT2A* fusions were the most common ($n = 28$, 60%, GRIN $p = 1.86 \times 10^{-74}$)(Fig. 3a, Supplementary Data 18–20, & Supplementary Fig. 9) and other in-frame fusions previously reported in myeloid malignancies involving *NUP98* ($n = 3$) and *ETV6* ($n = 2$) were also observed[32–34]. Likewise, 3 in-frame *RUNX1* fusions (*RUNX1-MTAP*, *RUNX1-LYPD5*, and *RUNX1-MECOM*) were identified (Supplementary Figs. 10 & 11). In addition to the *RUNX1-MECOM* fusion, we noted variable expression levels of *MECOM* across the cohort (FPKM range: 0.004–38.4), and 24 cases (43%) had an FPKM > 5 (*MECOM*[High]) (Fig. 3b). Elevated *MECOM* expression has been associated with myeloid neoplasms, particularly tMN and those with *KMT2Ar*, and is associated with a poor prognosis in both adult and pediatric myeloid neoplasms[34–39]. *KMT2Ar* was significantly enriched in the *MECOM*[high] cases (*KMT2Ar*: 18 vs. no *KMT2Ar*: 6, Fisher's $p < 0.01$) (Supplementary Fig. 12) while another *MECOM*[high] patient had a *NUP98* fusion (*NUP98-HHEX*)(Fig. 3b & Supplementary Fig. 10b), a previously reported association with high *MECOM* expression[40–42]. WGS on 3 of the 4 remaining *MECOM*[high] cases revealed structural variations (SV) involving the *MECOM* locus on chromosome 3 (Fig. 3c). Two cases involved noncoding regions of chromosome 2 adjacent to *ZFP36L2*, a gene encoding an RNA binding protein that is highly expressed in hematopoietic cells and is involved in hematopoiesis, and the other involved noncoding regions of chromosome 17 adjacent to *MSI2*, another gene encoding an RNA binding protein that has been found to be recurrently rearranged in hematological malignancies (Fig. 3d)[43–47]. The existing ENCODE data and similar studies in human CD34 cells support that these regions of the genome are super-enhancers in hematopoietic cells, suggesting a proximity effect in which these enhancers have been hijacked to drive high levels of *MECOM* expression (Supplementary Fig. 13)[48,49]. Furthermore, despite the lack of in-frame fusions in the RNA-seq data, these cases demonstrate allele-specific *MECOM* expression[50], further suggesting a cis-regulatory element may be driving this aberrant expression (Fig. 3d). WGS also identified a *MECOM* SV in SJ030441 (*SATB1@-MECOM*), but elevated *MECOM* RNA levels were not present in this case (Fig. 3b); however, immunohistochemical studies on the patient material demonstrated high *MECOM* protein expression in the blasts (Fig. 3e). Similar *MECOM* protein expression was detected in the other *MECOM* altered cases[51], but not in tMN cases without a *MECOM* SV (Fig. 3e). Contrary to pediatric de novo AML studies, there was not a statistically significant association between higher *MECOM* expression and disease-related deaths within this pediatric tMN cohort (Supplementary Fig. 14)[36]. Rather, a multivariable analysis shows that the presence of complex cytogenetics does significantly impact disease-related mortality risk (Fine-Gray model HR = 2.17; $p = 0.04$).

**Clonal evolution of pediatric tMN.** Finally, using a combination of targeted capture resequencing and a bioinformatic error suppression approach[52] we described the timing of acquisition and evolution of the somatic mutations for 37 cases using samples from interval time points prior to the development of tMN, including 26 cases in which material for the primary malignancy was available for analysis (Supplementary Data 21). We demonstrated that the somatic variants most commonly arose after the introduction of cytotoxic therapy ($n = 23$ of 26, 88%), and we

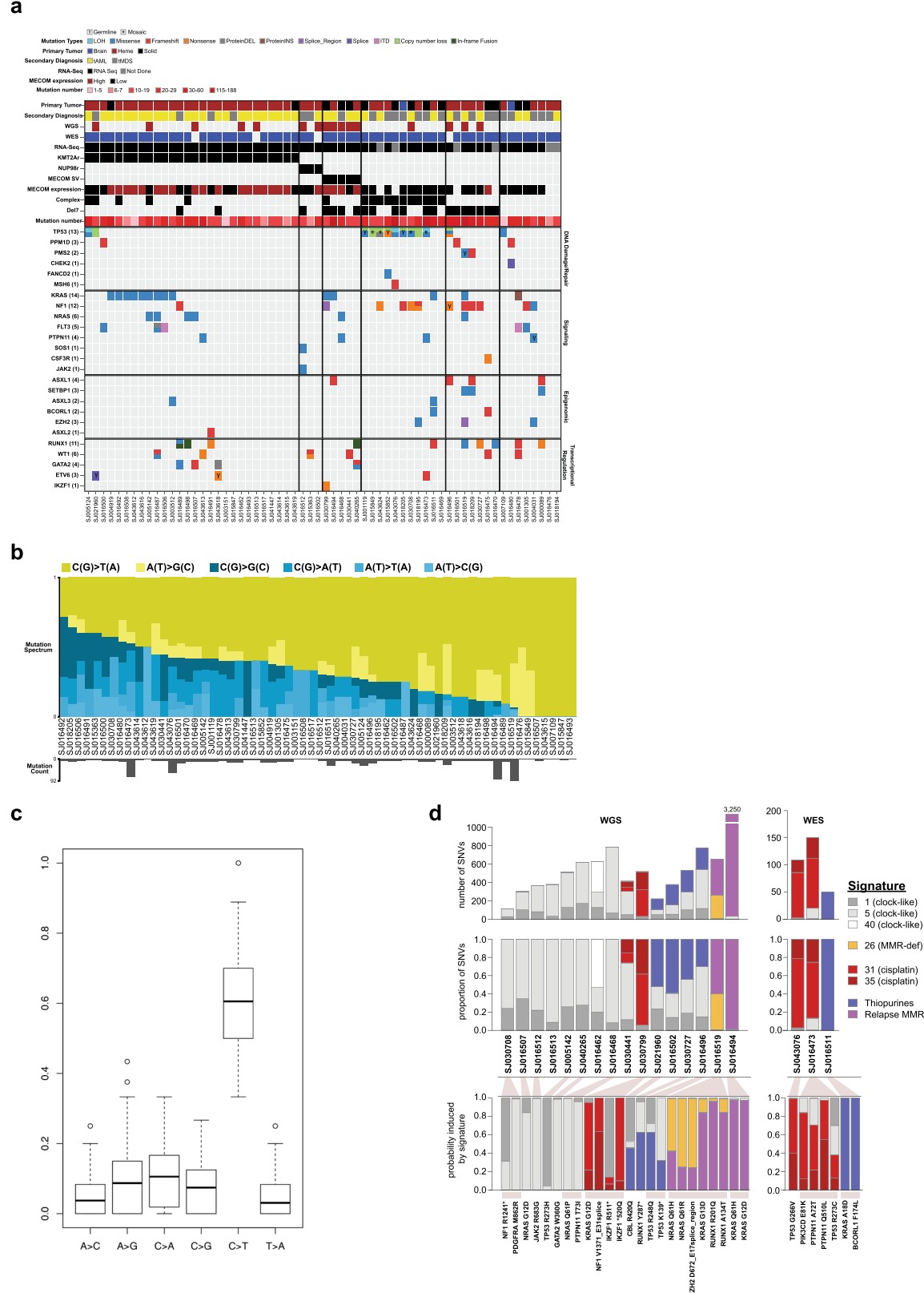

could detect these acquired mutations up to 748 days (mean: 405 days; range: 118–748) prior to morphologic evidence of tMN (Fig. 4a & Supplementary Figs. 15 & 16). Three cases were found to be clonally related to the original malignancy. These included a tMDS that developed 8 months after AML and both were found

to harbor a *NUP98-NSD1* fusion (Fig. 4b) with multiple discrete *WT1^mut* subclones, and 2 cases where the initial lymphoid malignancy (ALL or NHL) and tMN developed from a common clone that subsequently underwent a lineage switch (Fig. 4c–f). Unlike adult tMN[5], the somatic *TP53* variants could not be

**Fig. 2 Comprehensive mutational spectrum of pediatric tMN. a** Heat map showing the integrated analysis of the pediatric tMN cohort with tumor and non-tumor material ($n = 62$). **b** Mutational spectrum of 62 tumor/normal pairs. Yellow and blue bars show the relative contribution of transitions and transversion. Gray bars at bottom indicate number of mutations present for each patient. **c** Bar graph showing the mean relative contribution of each transition or transversion. C > T transitions are the most common transition or transversion in 60 of 62 patients (96.7%; 95% CI: 88.8–99.6%; $p = 2.7 \times 10^{-44}$ by exact binomial test). Boxes delineate the upper and lower quartiles and the black bar indicates the median. **d** Mutation signature analysis on 16 cases with available WGS and 3 cases with WES with >30 SNVs. Top: absolute number of SNVs and the contribution of specific COSMIC, thiopurine, and relapse MMR signatures. Middle: relative contribution of specific COSMIC, thiopurine, and relapse MMR signatures. Bottom: select disease relevant mutations present in each patient and the probability that each is induced by the indicated mutational process.

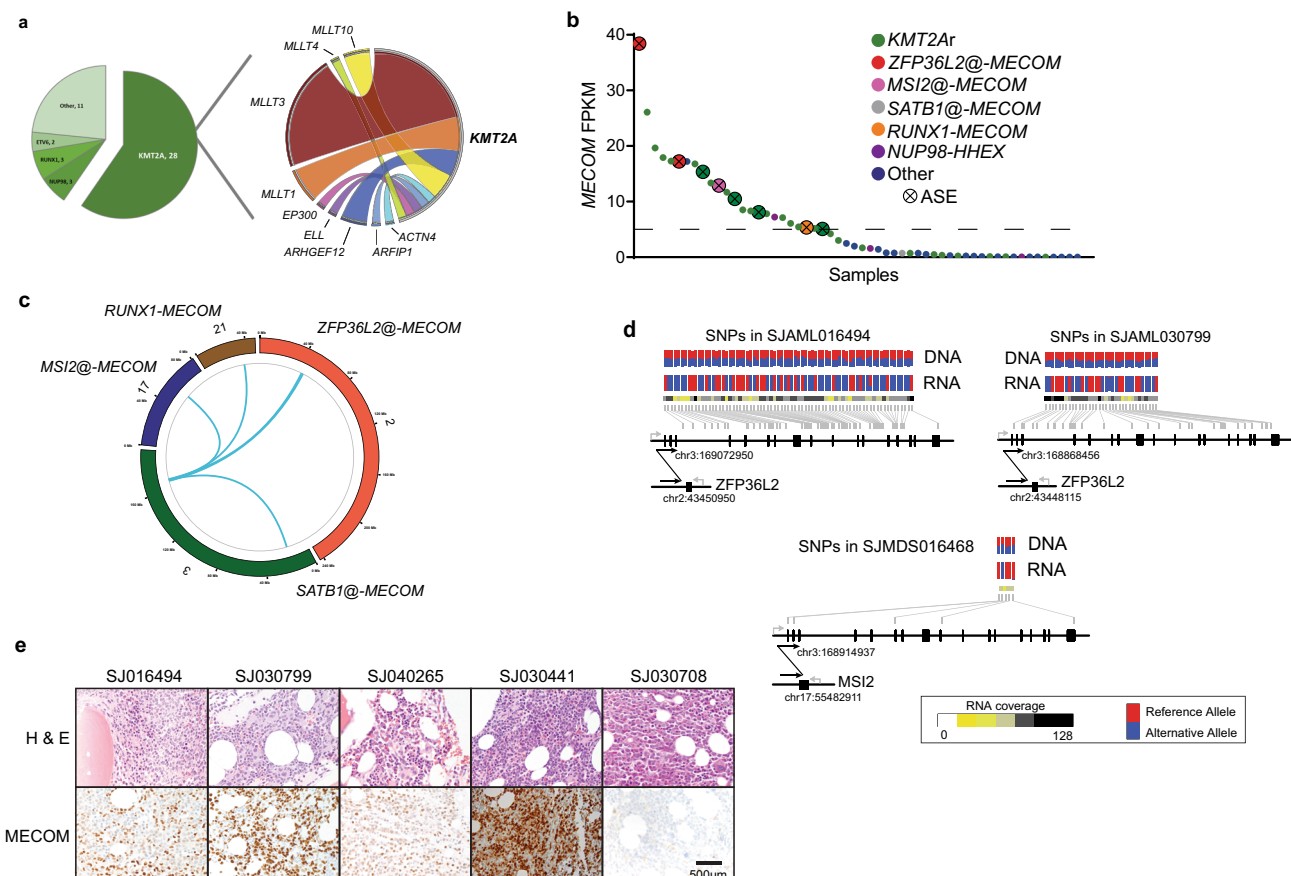

**Fig. 3 Structural variations and *MECOM* dysregulation in pediatric tMN. a** Pie chart showing the distribution of in-frame fusions ($n = 47$) found in the pediatric tMN cohort (left). Ribbon plot showing the *KMT2A* binding partners found in pediatric tMN (right). The weight of the ribbon correlates to the frequency of the fusion. **b** *MECOM* FPKM plot for cases with RNA-Seq ($n = 56$). Dashed line indicates the level above which cases were classified as *MECOM*^High. ASE allele specific expression. **c** Circos plot indicting the MECOM SVs found in the pediatric tMN cohort. Chromosome number and specific SV is listed around outside of ring. **d** Allele-specific RNA expression resulting from structural variants[50]. Heterozygous SNPs (genomic positions indicated by gray lines; red: reference allele; blue: alternative allele) detected in tumor DNA exhibited mono-allelic expression in tumor RNA. Structural alterations are indicated by arrows with breakpoints listed. Sequencing depth for each SNP in RNA-Seq are indicated as a heatmap. **e** Photomicrographs of bone marrow core biopsy of 4 cases with high *MECOM* expression (right panels: MECOM (Evi-1) IHC: 1C50E12, Cell Signaling Technology, dilution: 1:500) and a control case (SJ030708) with low/absent *MECOM* expression. Immunohistochemistry was performed once on the patient material available. All images are at equal magnification (20x).

detected with ultra-deep amplicon sequencing (72,000x) and bioinformatic error suppression in pre-treatment samples[52] (Supplementary Data 22 & Supplementary Fig. 17).

## Discussion

Here we show the results of our comprehensive sequencing of pediatric tMN which reveals that *KMT2A*r are the most common driver alterations in our pediatric tMN cohort along with Ras/MAPK pathway mutations. Somatic *TP53* alterations were also frequent, but these mutations appeared to arise after chemotherapy, unlike adult tMN[5]. Additionally, we identified *MECOM* overexpression to be frequent, and in some of these cases the overexpression was driven by enhancer hijacking. Finally, we show that pediatric tMN-defining variants arise most commonly as a consequence of cytotoxic therapy, and that these malignant clones can be identified, on average, >1 year before morphologic evidence of neoplasm. While these studies reflect the experience of a single institution, the findings highlight the diverse nature of genomic alterations in pediatric tMN and suggest that genomic screening approaches may be able to identify at risk patients prior to tMN development.

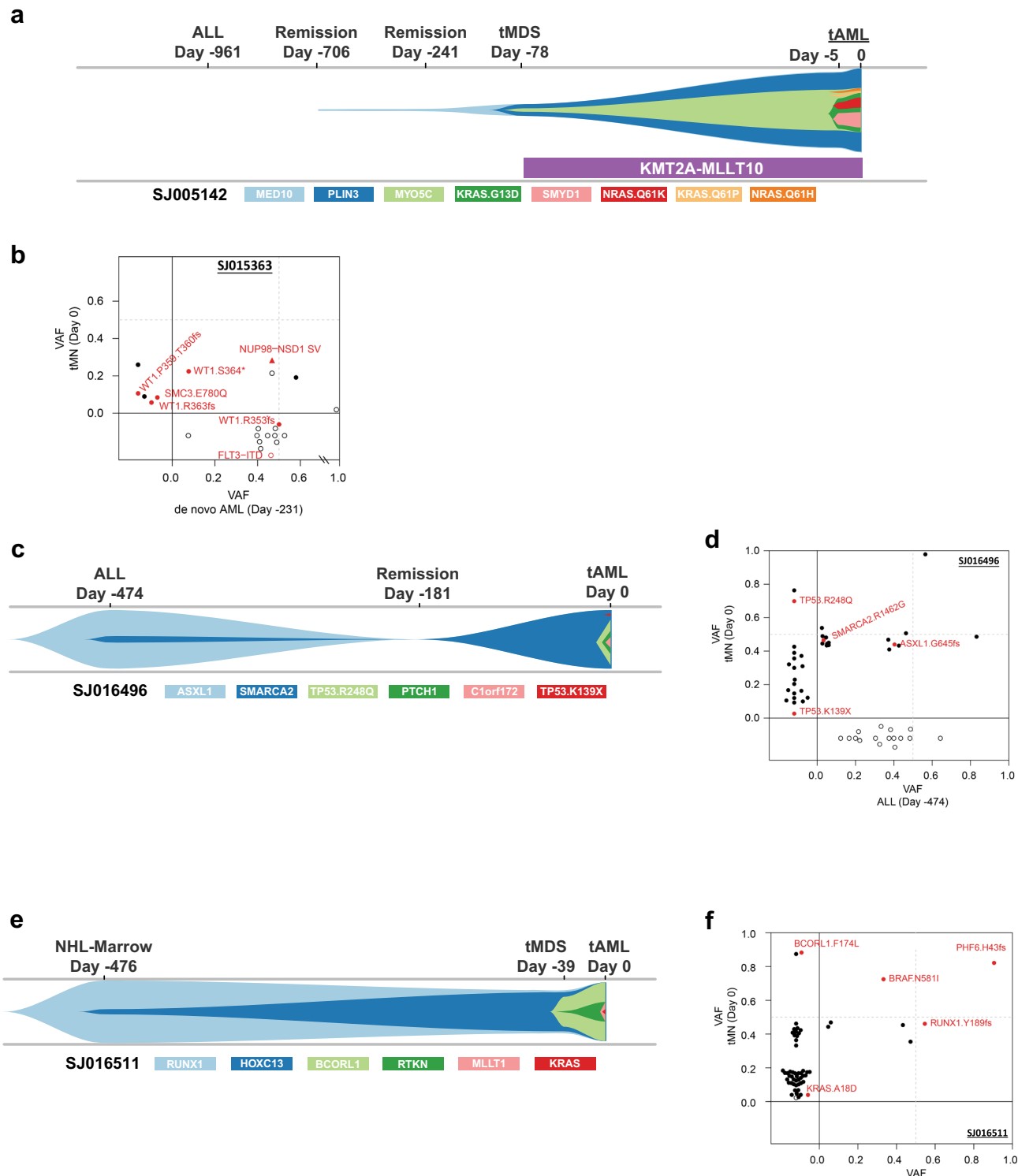

**Fig. 4 Clonal evolution of pediatric tMN. a** A river plot showing a representative case where tMN variants occurred only after exposure to cytotoxic therapy. In this case the founding tMN clone was detectable 628 days prior to morphologic diagnosis of tMDS. **b** A 2-dimensional VAF plot showing that the tMN and de novo AML were actually related via a *NUP98-NDS1* fusion (red triangle) and a subclonal *WT1* variant. **c**, **d** River- and 2d-plots showing an ALL related to the subsequent tMN through an *ASXL1*-mutant founding clone with a *SMARCA2* subclone, and following chemotherapy an outgrowth of the *SMARCA2* clone with subsequent acquisition of 2 *TP53* subclones. **e**, **f** River- and 2d-plots showing staging bone marrow collected at time of NHL diagnosis related to the subsequent tMN through a *RUNX1* founding clone with eventual acquisition *BCORL1* and *KRAS* subclones, which paralleled the development of tMDS and tAML, respectively. 2-d plot NOTE: upper right-hand quadrant contains shared variants between the 2 time-points (*X* and *Y* axes). Open symbols indicate variants with WGS or WES only. Closed symbols indicate variants validated via capture resequencing.

## Methods

**Patient sample details.** Patient material was obtained with written informed consent using a protocol approved by the St. Jude Children's Research Hospital Institutional Review Board. All patients with a diagnosis of tMN (either tMDS or tAML) with appropriate consent for genomic studies and available tumor or normal samples banked in the St. Jude Tissue Biorepository were included. Diagnoses were reviewed by a hematopathologist (J.M.K.) and classified according to the WHO 2016 classification of myeloid neoplasms and acute leukemia[53]. Supplementary Data 1 contains clinicopathological information for all samples included in our analyses. Samples were de-identified before nucleic acid extraction and analysis. The study cohort is comprised of 84 total patients (tMDS = 28, tAML = 56). Sixty-two patients had available tumor and normal tissue for characterization, while the remaining 22 lacked sufficient tumor material for comprehensive sequencing (Table 1). For the 62 tumor/normal pairs, flow sorted lymphocytes from the diagnostic tMN samples were used as the source of normal comparator genomic DNA in 53 cases, while bone marrow (n = 4) or peripheral blood (n = 5) from alternate timepoints was used for the remainder. Cryopreserved bulk bone marrow cells were thawed in a 37 °C water bath and transferred to 20% FBS in PBS to remove residual DMSO according to standard approaches[54]. Cells were lysed with ACK lysing buffer (ThermoFisher A1049201) and washed with PBS prior to staining. The following antibodies were used to immunophenotype the cells and facilitate flow sorting of myeloid and lymphoid populations: CD15-FITC (eBioscience, clone HI98), CD71-BV711 (BD Biosciences, clone M-A712), CD34-PE (Beckman, clones QBEnd10, Immu133, Immu409), CD45R-PerCP-Cy5.5 (eBioscience, clone RA3-6B2), CD235a-PE-Cy7 (BD Biosciences, clone GA-R2), CD3-APC-Cy7 (BD Biosciences, clone SK7), CD33-APC (eBioscience, clone WM-53). For the 23 normal only cases, bulk sequencing was completed on interval remission samples.

**WGS, WES, and RNA-Seq analysis.** DNA and RNA material was isolated from bulk myeloid or isolated lymphocytes by standard phenol:chloroform extraction and ethanol precipitation. Whole genome sequencing libraries were constructed using the TruSeq DNA PCR-Free sample preparation kit (Illumina, Inc., CA) following the manufacturer's instructions and whole-exome sequencing was completed using the Nextera Rapid Capture Expanded Exome reagent (Illumina). After library quality and quantity assessment, WGS, WES, or RNASeq samples were sequenced on various Illumina platforms (HiSeq 2500, HiSeq 4000, or NovaSeq 6000). Mapping, coverage, quality assessment, single-nucleotide variant (SNV) and indel detection, and tier annotation for sequence mutations (SNVs discovered by WGS were classified as tier 1, tier 2, tier 3, or tier 4) have been described previously[55–57] and briefly described here. DNA reads were mapped using BWA[58,59] (WGS: v0.7.15-r1140; WES: v0.5.9-r26-dev and v0.7.12-r1039 since data was generated over a period of time) to the GRCh37/hg19 human genome assembly. Aligned files were merged, sorted and de-duplicated using Picard tools 1.65 (broadinstitute.github.io/picard/). SNVs and Indels in WGS and WES were detected using Bambino[60]. For WGS data, sequence variants were classified into the following four tiers: (i) tier 1: coding synonymous, nonsynonymous, splice-site and noncoding RNA variants; (ii) tier 2: conserved variants (conservation score cutoff of greater than or equal to 500, based on either the phastConsElements28way table or the phastConsElements17way table from the UCSC Genome Browser) and variants in regulatory regions annotated by UCSC (regulatory annotations included are targetScanS, ORegAnno, tfbsConsSites, vistaEnhancers, eponine, firstEF, L1 TAF1 Valid, Poly(A), switchDbTss, encodeU-ViennaRnaz, laminB1 and cpgIslandExt); (iii) tier 3: variants in non-repeat masked regions; and (iv) tier 4: the remaining SNVs. Structural variations in whole-genome sequencing data were analyzed using CREST[61] (v1.0). RNA-sequencing was performed using TruSeq Stranded Total RNA library kit (Illumina) and analyzed, as previously described[16,17]. Briefly, RNA reads were mapped using our StrongARM pipeline (internal pipeline, described by Wu et al.[62]). Paired-end reads from RNA-seq were aligned to the following four database files using BWA: (i) the human GRCh37-lite reference sequence, (ii) RefSeq, (iii) a sequence file representing all possible combinations of non-sequential pairs in RefSeq exons and, (iv) the Ace-View database flat file downloaded from UCSC representing transcripts constructed from human ESTs. Additionally, they were mapped to the human GRCh37-lite reference sequence using STAR. The mapping results from databases (ii)–(iv) were aligned to human reference genome coordinates. The final BAM file was constructed by selecting the best of the five alignments. Chimeric fusion detection was carried out using CICERO[63] (v0.3.0) and Chimerascan[64] (v0.4.5). All identified fusions were validated by either RT-PCR, cytogenetics, manual review of CREST data, or a combination of these methods (Supplementary Data 18, 20, & Supplementary Figs. 9 and 18). Mapping statistics and coverage data are described in Supplementary Data 6–8 & 15. Recurrent SNV's identified by WGS or WES were validated by custom capture resequencing (Supplementary Data 2, 3, and 19). Custom capture baits were designed (Twist Biosciences) to be 80 nucleotides long covering the provided hg19 target region consisting of 1,006,633 unique base pairs (bp). A total target region of 904,622 bp is directly covered by 11,455 probes. BWA[58,59] (v0.7.12) MEM algorithm was used to map the TWIST sequencing reads to the GRCh37/hg19 human genome assembly. Rsamtools[65] (v1.30.0) was used to retrieve read counts from BAM files for the SNV/Indels called in WES, requiring MAPQ > = 1 and base quality Phred score > = 20. We also performed de novo

mutation calling in an attempt to catch canonical low variant allele frequency (VAF) cancer gene mutations missed by WES using VarScan[66] (v2.3.5) on the TWIST data with the following criteria: MAPQ > = 1; base quality Phred score > = 20; VAF > = 0.01 and variant call p-value < = 0.05. Selected somatic variants (WES read count <5 and targeted capture read count <10) and all somatic TP53 variants identified via WES were validated by custom amplicon sequencing. PCR primers (Supplementary Data 22) were designed to flank the putative variants. Amplicon sizes were approximately 200 base pairs. PCR was performed using KAPA HiFi HotStart ReadyMix (Roche), 100 nM of each primer (IDT) and 20 ng of gDNA in a 40uL reaction volume. Thermocycling was performed using the following parameters: 95 °C for 3 min; 98 °C for 20 s, 62 °C for 15 s, and 72 °C for 15 s for a total of 30 cycles; and 72 °C for 1 min. All amplicons were quality checked on a 2% agarose gel. Primers were designed to incorporate Illumina overhang adapter sequences which allowed for indexing using the Nextera XT Index kit (Illumina) following the manufacturer's instructions. Libraries were normalized, pooled, and sequenced on an Illumina MiSeq instrument using a 2 × 150 paired-end version 2 sequencing kit. We used the CleanDeepSeq[52] approach with default settings for error suppression in this ultra-deep amplicon sequencing.

**Copy number analysis using NGS data.** Copy number analysis of the WGS (n = 4) cases was done using CONSERTING[67]. Copy number analysis of the WES (n = 58) cases was done following these steps: Samtools[68] (v1.2) mpileup command was used to generate an mpileup file from matched normal and tumor BAM files with duplicates removed; VarScan2[66] (v2.3.5) was then used to take the mpileup file to call somatic CNAs after adjusting for normal/tumor sample read coverage depth and GC content; Circular Binary Segmentation algorithm[69] implemented in the DNAcopy R package[70] was used to identify the candidate CNAs for each sample; B-allele frequency info for all high quality dbSNPs heterozygous in the germline sample was also used to assess allele imbalance.

**Germline analysis.** Whole exome sequencing data were analyzed using internal workflows that were previously described[19]. Briefly, the sequencing data were analyzed for the presence of single-nucleotide variants and small insertions and deletions (Indels) and for evidence of germline mosaicism. Germline copy-number variations and structural variations were identified with the use of the Copy Number Segmentation by Regression Tree in Next Generation Sequencing (CONSERTING)[67] and Clipping Reveals Structure (CREST)[61] algorithms. For all SNPs and Indels, functional prediction (e.g., SIFT, CADD, and Polyphen) scores and population minor allele frequency (MAF) were annotated. In this work, 3 databases were used for population MAF annotation: (i) NHLBI GO Exome Sequencing Project (http://evs.gs.washington.edu/EVS/); (ii) 1000 genomes (http://www.internationalgenome.org); and (iii) ExAC non-TCGA version (http://exac.broadinstitute.org/). For missense mutations, REVEL (rare exome variant ensemble learner) score was also determined to help predict pathogenicity[71]. A gene list of 631 genes were composed from various resources: (i) literature review of genes that are potentially involved in AML, MDS, inherited bone marrow failure syndromes, as well as other cancer types[5,19,72–74] (ii) genes that were involved in splicing from predefined pathways (e.g., splicing) in KEGG, GeneOntology, Reactome, Gene Set Enrichment Analysis (GSEA), and NCBI (Supplementary Data 14). The following filtering criteria were applied: VAF ≥ 0.2, coverage >20x, ExAC MAF < 0.001 (or not present in ExAC), REVEL score >0.5 (for missense mutations), NHLBI and 1000 genomes MAF < 0.001. One TP53 variant that was lost through this filtering was manually recovered because the patient was clinically diagnosed with Li Fraumeni syndrome. Given this finding, all germline TP53 mutations were manually reviewed and analyzed as described below for mosaicism. Of note, the germline ETV6 p.N386fs in case SJ021960 was previously reported[75]. All non-synonymous mutations were comprehensively reviewed and classified as pathogenic, likely pathogenic, of uncertain significance, likely benign, or benign based on recommendations from the American College of Medical Genetics and Genomics and the Association for Molecular Pathology[76] by members of the Cancer Predisposition Division at St. Jude (J.L.M and K.E.N).

**Determination of mosaicism versus tumor-in-normal contamination.** Because the normal samples used were hematopoietic specimens (sorted lymphocytes or remission bulk marrow), the mosaic mutations can be a result of incomplete remission. To rule out this possibility, we performed a previously developed statistical analysis that can model residual disease burden[19]. Briefly, we first determined purity (denoted as f) of the tMN tumor sample by clustering allele fractions of somatic SNVs/Indels by using R package "Mclust," where the cluster with the highest mean (denoted as u) center under 0.5 was used to estimate tumor purity (multiplied by 2 to account for diploid status, $f = 2*u$). To account for clonal evolution, we also calculated tumor purity by using heterozygous loss and copy neutral loss of heterozygosity (CN-LOH) regions with the highest magnitude of scores. For heterozygous loss regions, the purity is estimated as $f = 2 - 2^{(\log.\text{ratio}+1)}$, while for CN-LOH region the purity is estimated as $f = 2*AI$ where $AI = |$ B-allele fraction $-0.5|$. The maximum of the SNV/Indel and CNV/LOH-based purity estimate was used as the final purity estimate (f) for a given tumor. We then defined an SNV/Indel as diploid clonal if its allele fraction is $> f*0.5*80\% = u*80\%$ and <0.6. The sum of mutant allele counts of these markers was denoted as M, and

the sum of depth of these markers as $T$, thus the tumor-in-normal contamination level of the germline sample is then estimated as $c = M/T$. The expected allele fraction of $TP53$ mutation is estimated by considering its local ploidy and contamination level c. In our dataset, the $TP53$ mutations are either 1-copy loss-LOH or CN-LOH (Supplementary Data 1, 4, and 16). For 1-copy-LOH, the expected allele fraction of $TP53$ under contamination is $e = c*(2-c)^{-1}$, while for CN-LOH the expected allele fraction of $TP53$ is simply $e = c$. We then tested the hypothesis that the observed $TP53$ allele counts in germline sample are due to contamination by using a binomial test. A significant $p$ value (<0.01), after Bonferroni correction, would indicate that the observed allele counts are unlikely to be explained by contamination. To rule out the possibility of germline inheritance, we also tested the allele counts against inheritance (i.e., $e = 0.5$). A $TP53$ mutation with significant $p$ values (<0.01) for both the contamination test and the inheritance test is called a mosaic mutation. For normal only samples, variants with a VAF of ≥0.2 were classified as germline, but variants with a VAF of <0.2 and with a supportive clinical history were classified as mosaic. We are unable to distinguish germline versus somatic mosaicism.

**Mutational signature analysis.** The trinucleotide context of each somatic SNV was identified using an in-house script, and mutations were assigned to one of each of the 96 trinucleotide mutation types[77]. To detect whether any novel signatures were present in the dataset, we ran SigProfiler version 2.3.1[78] on the SNV catalogs from the 16 WGS samples and extracted 3 signatures. One of the extracted signatures resembled the cisplatin signature (SBS-31); one represented a combination of clock-like signatures 1 and 5 (SBS-1, SBS-5)[77], and the third resembled a signature recently reported in relapsed ALL of unknown cause which was only present in patients with germline or somatic $PMS2$ alterations. This third signature (termed the "relapse MMR" signature) was also similar to the thiopurine signature we recently reported[28], with similar strand bias, and is potentially therefore a modified thiopurine signature in samples with MMR defects. We tested for the presence of the 60+ COSMIC v3 signatures in each WGS sample using SigProfilerSingleSample (version 1.3) and the COSMIC v3 signature definitions provided with that version of the software. From this analysis, signatures never exceeding 150 mutations in any one sample were identified and excluded from our final analysis in order to avoid likely spurious signatures. Based on these data, our finalized WGS signature data were obtained by testing for the presence of only the following signatures in each sample using SigProfilerSingleSample: COSMIC signatures 1, 5, and 40 (clock-like), COSMIC signature 26 (MMR deficiency), COSMIC signatures 31 and 35 (cisplatin), the experimental thiopurine signature we recently reported, generated by treating MCF10A cells with thioguanine[28], and the relapse MMR signature. We used a required cosine increase of 0.02 or more for a signature to be detected in a single sample, and default parameters otherwise. For exome samples, we likewise tested for these signatures using SigProfilerSingleSample, but excluded from our analysis exome samples that had cosine reconstruction scores of less than 0.9 (comparing the sample's SNV catalog profile with the profile as reconstructed by signatures) or less than 30 SNVs total, or which already had WGS data, resulting in only 3 exome samples with usable signature data. We calculated the probability that individual SNVs were caused by a signature as done by others[79] and as we reported previously[28]. The probability that a variant was caused by a specific signature was calculated as follows. Let $s_k$ represent the signature strength vector for a given sample (measured in number of SNVs caused by the signature), where $k = 1, 2, …, 8$ is one of 8 signatures we identified, such that $s_1$ equals the number of specific SNVs caused by signature 1 in the sample, and $\Sigma s_k$ equals the total number of SNVs in the sample. Let $c = 1, 2, …, 96$ represent each of the 96 possible trinucleotide mutation types. Each of the $k$ signatures mutates each of these 96 trinucleotide mutation types $c$ with a probability $P_{c,k}$ (ranging from 0 to 1.0) where the sum of the probabilities for a given signature across all 96 trinucleotide mutation types is 1.0. The probability that a mutation of interest $m$ (at trinucleotide mutation type $c$) was caused by a specific signature $i$ is calculated as shown in Eq. 1:

$$P(i|m) = \frac{S_i^* P_{c,i}}{\sum_{k=1}^{11} \left( S_k^* P_{c,k} \right)} \quad (1)$$

**GRIN analysis.** The genomic random interval (GRIN) method[18] was used to evaluate the statistical significance for the prevalence of SNVs, heterozygous deletions, fusion breakpoints, copy-neutral loss-of-heterozygosity, and amplification in each gene. For each gene, a $p$-value for each of these genomic alterations was computed. Also, for each gene, an overall $p$-value was computed by finding the minimum $p$-value across the five lesion types and comparing it to the beta distribution corresponding to the distribution of the minimum of five id uniform (0,1) realizations. For each set of $p$-values (one for each lesion type and the overall $p$-value), a robust method[80] was used to compute false discovery rate estimates, which are reported with the symbol $q$. A total of 91 genes were identified as statistically significant with an overall $q < 0.05$. Additionally, MutSigCV[81] analysis was used to determine driver status of SNVs and indels.

**Super enhancer analysis in CD34+ cells.** H3K27ac ChIP-seq data were downloaded from GEO accession GSE104579[82]. Raw reads were adapter-trimmed and subject to quality filtering using Trim Galore (v0.4.4), retaining reads with a quality

score >20. Reads were mapped to the human genome (GRCh37) using BWA (v0.7.12)[58], converted to bam format, and duplicate reads were marked using bio-bambam2 (v2.0.87)[83] and removed using samtools (v1.10)[68]. H3K27ac peaks were called using macs2 (v2.1.1)[84] in BEDPE mode with a $p$-value cutoff of $1 \times 10^{-5}$. ROSE was run using the de-duplicated H3K27ac and input bam files and the macs2 peak file with default parameters. For additional visualization of the chromatin landscape in human CD34 + cells, three additional datasets were included in IGV snapshots. The CTCF bigwig file was downloaded from GEO accession GSE104579. The "CD34 + H3K27ac (Roadmap)" wiggle file was downloaded from GEO accession GSM772885[85] and converted to bigwig. CD34+ ATAC-seq data were downloaded from GEO accession GSE74912[86] and all biological replicates for CD34+ samples were merged into a single bedGraph file and converted to bigwig format for visualization. All RNA-seq tracks are normalized read coverage.

**Statistical methods.** The Wilcoxon–Mann–Whitney non-parametric test, two-tailed, was used to compare means of quantitative variables across two experimental groups or diagnostic groups. The Fisher's exact test was used to compare the frequency of complex karyotype between patients with and without $TP53$ mutations. Survival analysis of cause-specific death was performed with a Fine-Gray model[87] that accounts for different causes of death as competing events and adjusts for hematopoietic stem cell transplant as a time-dependent outcome predictor variable.

**Reporting summary.** Further information on research design is available in the Nature Research Reporting Summary linked to this article.

## Data availability

The genomic data generated in this study have been deposited in the European Genome-Phenome Archive (EGA), which is hosted by the European Bioinformatics Institute (EBI), under accession EGAS00001004850 and through St. Jude Cloud [https://pecan.stjude.cloud/permalink/tMN]. All other remaining data are available within the article and supplementary files or available from the authors upon request. Other publicly available datasets used for CD34+ cell super-enhancer analysis are deposited in Gene Expression Omnibus (GEO): H3K27ac and CTCF ChIP-seq data are available under accession number GSE104579, CD34 + H3K27ac Roadmap ChIP-seq data are available under accession number GSM772885, and CD34+ ATAC-seq data are available under accession number GSE74912.

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

## Acknowledgements

We thank all the patients and their families at St. Jude Children's Research Hospital (SJCRH) for their contribution of biological specimens used in this study. We also thank the Biorepository, the Flow Cytometry and Cell Sorting Core, and the Hartwell Center for Bioinformatics and Biotechnology at SJCRH for their essential services. Julie Justice in the Anatomic Pathology lab established the immunohistochemistry for MECOM. J.R.S. is supported by the NHLBI (1K08HL150282-01) and Alex's Lemonade Stand Foundation Young Investigator Award. This work was funded by the American Lebanese and Syrian Associated Charities of St. Jude Children's Research Hospital and grants from the US National Institutes of Health (P30 CA021765, Cancer Center Support Grant; R01 HL144653 to J.M.K.). J.M.K. holds a Career Award for Medical Scientists from the Burroughs Wellcome Fund. Support was also provided by the Edward P. Evans Foundation (J.M.K.).

This research content is solely the responsibility of the authors and does not necessarily represent the official views of the National Institutes of Health.

## Author contributions

J.R.S., J.M., J.K., S.W.B., L.M., X.M., and J.M.K. prepared the manuscript. J.R.S., T.W., R.H., J.K., T.G., X.M., and J.M.K. were responsible for experimental design and analysis. T.W. prepared DNA and RNA from all patient samples. J.M., M.P.W., J.R.M., X.C., G.S., G.W., Y.L., J.E., S.N, and J.Z. were responsible for bioinformatic data analysis. L.M. performed the super-enhancer analysis of CD34$^+$ cells. K.E.N., M.F.W., J.L.M., J.K., J.R.S., T.G., and J.M.K. analyzed germline variants and determined their likely pathogenicity. S.W.B. performed and analyzed the mutational signatures present in the tMN cohort. J.R.S., J.K., and C.B. assembled all clinical data for the tMN cohort. P.K. performed MECOM immunohistochemistry. S.P. and H.W were responsible for all statistical analyses. C.G.M. and J.E.R. assisted with data analysis and acquisition of patient cases.

## Competing interests

The authors declare no competing interests.
