## [Peer Review File · Nature Communications]

REVIEWER COMMENTS

Reviewer #1 (Remarks to the Author): Expert in tMNs and bioinformatics

Supplementary Figure 1. This distribution looks bimodal. Can you fit a two-hump distribution to it and give separate estimates of the two peaks? Does anything correlate with being in the second peak (at larger times)? I'm wondering if the gene expression pattern in the second peak supports there being more of an immune response against the clone, i.e. was this maybe the mechanism of delaying onsets of these t-MN? It would also be good to know how many background cases of t-MN were expected in the parental cohort of all children treated for 1st cancers, i.e. might these late arrivals instead really be coincidental background MN cases rather than true t-MN? The expected number needs to be calculated based on background incidence rates of 1st MN obtained from SEER and person years at risk, i.e. survival times of subjects within the much larger (~100 to 200-fold) parental cohort from which these 85 t-MN cases arose. If the number is less than 1, I'd like to know if the immune system caused the delays. If it is greater than say 10, then late arrivals are more likely to be background cases, and those with longer survival times are also more likely to be background cases (since true t-MN have worse survival), and it might be nice to assign probabilities of being background cases to each person and seeing if that correlates with anything.

Supplemental Figures 7 and 10A. The resolution here is bad. These figures should be made as pdfs (i.e. vector graphics) and read into Word as a file, not pasted in via the clipboard, as this converts them into bitmaps. Doing this will also reduce the file size.

Sup Tables 2, 3, 5, and 7-11 need a freeze pane to hold the header at the top of the excel sheet.

Tomas Radivoyevitch

Reviewer #2 (Remarks to the Author): Expert in paediatric and juvenile myeloid neoplasms, de novo and treatment-related

Schwartz and colleagues present a manuscript titled "The Acquisition of Molecular Drivers in Pediatric Therapy-Related Myeloid Neoplasms". They studied a cohort of 62 patients with therapy related myeloid neoplasms (tMN) in both tumor and non-tumor material using WGS, WES, and RNAseq. 23 cases with tMN were profiled only in non-tumor specimens using WES. Common alterations included mutations in Ras pathway, RUNX1, and TP53, and KMT2A rearrangements (in 28/56, 60%) and high MECOM expression (24/56, 43%). Patients with high MECOM expression had either KMT2A rearrangements or NUP98 fusions. Most of the mutations found in tMN were not found in primary tumors. Germline alterations with pathogenic/likely pathogenic effect were identified in 13/85 (15%) patients. While this work has technical merit and brings novel aspects to the understanding of genetics of tMN, some of the statements are misleading or inaccurate, and the manuscript is somewhat confusing.

Major comments:

1. Many findings are well known and have previously been reported, for example that complex cytogenetics is associated with poor survival, or that the tMN population is older than that of primary MDS. Also, MECOM locus aberrations were previously reported in tMN (PMID 22706870) in a cohort with a pediatric patient. The authors should comment on that.

2. The authors write that the prevalence of germline mutations found in this cohort (13/85, 15%) was higher than what has been previously reported for childhood cancers. This statement is misleading and “inflating the numbers”, because many of the genes shown to be mutated (Table 2) are not associated with cancers. Examples: FANCD2 (homozygous FA, heterozygous healthy carrier) and PMS2 (homozygous CMMRD, heterozygous Lynch syndrome). Other genes shown are dubious (ARID2, EZH2, CREBBP, RPL22, TRIP11). They might be associated with developmental syndromes, but are not associated with cancer. Most likely these genes were selected by the authors because they are associated with somatic changes in cancer genomes. To avoid misleading statements, the authors should report prevalence based on well-established cancer predisposition syndromes (TP53, ETV6, NF1, PTPN11), and confirm the germline status in cases with no family history/phenotype. For example patient SJ016473 with TP53 mutation had a VAF of 7%. Was it confirmed in germline? Finally, table 6 includes 20 patients with germline mutations, but the authors write in text that 13 cases had germline mutations.

3. The cohort consisted of 85 patients with tMN following hematopoietic malignancies in 60 cases, solid tumors in 23 cases, and brain tumors in 2 cases. The majority of patients (50) had ALL as primary diagnosis. This is a bias for the whole study and should be acknowledged, especially when discussing results on KMT2CA translocations, and potentially MECOM expression. The authors should tone down the generalizing statements on “identifying molecular drivers in pediatric tMN” and instead clearly state that this is a single center experience with high prevalence of ALL cases, and discuss how etoposide treatment could have confounded the results. It would make sense to compare the somatic profiles based on chemotherapy (etoposide yes/no).

Other comments:

4. In many patients, the number of mutations is relatively few. These numbers may not be sufficient enough to describe the significance of C>T transition among transitions and transversions (Figures 2b and 2c). Did the authors also look for the relative contribution of each transition and transversions in 16 WGS data?

5. It is not clear how the authors correlate KRAS G12D variant to cisplatin signature in SJ030799 while the same variant in the patient SJ016494 was assigned to relapse MMR signature (Figure 2d).

6. The authors argue the monoallelic expression of MECOM as the supporting evidence for the cis-regulatory elements driving (enhancer high jacking) the increased levels of MECOM in ZFP36L2@-MECOM and MSI2@-MECOM. However, monoallelic expression was also observed in 4 KMT2Ar cases (Figure 3b).

7. Supplementary methods: “Determination of mosaicism versus tumor contamination” This is confusing to a Reader. Was the mutation analysis performed in germline material or does this method refer only to analyzing hematopoietic specimens?

8. There are several cases with very high mutational burden. Did the authors observe chromothripsis in their study cohort?

9. The authors write that deletions or CN-LOH involving chromosome 7 (del(7)) were found in 22 of 62 patients (35%)". However, in Supp. Table 1, 22 of 80 patients had such lesions.

10. Supp. Table 2 (somatic) and Table 9 (germline): it is not clear what material was used as germline control.

11. Supp. Figure 4b: based on tumor VAF the authors suggest that TP53 are early and PPM1D mutations are late, but no additional data is provided to support this statement. I suggest verifying this using single cell studies, or at least longitudinal studies.

12. Supp. Figure 8: the bottom gel is not clear. What primers were used, what length was expected? Were the products sequenced? I suggest repeating this experiment and sequencing the products to confirm the fusion sequence.

Minor comments:

1. Line 93, pathologic or likely pathologic can be written as pathogenic or likely pathogenic (these are most widely used terms)
2. Figures 3d,4c, and 4d are not referred to in the manuscript.
3. Supplementary figure 7- x-axis is not readable.
4. In the supplementary figures 3, 4 and 9, the term secondary protein structure is used. This has to be corrected as protein domain architecture.
5. A number of typos and incomplete figure legends. For example what is "*" and "Y" in Supp Fig 3d
6. Many unclear abbreviations in Supp. Table 17
7. Fig. 3e: SJ030708 panel is not visible
8. What does the finding of C>T transitions mean? This is a known mutational signature of UV light / epithelial cancers. The authors should discuss this.

Reviewer #3 (Remarks to the Author): expert in leukaemia genetics and leukaemogenesis

In this study, the authors performed comprehensive analyses of genomic mutations in pediatric therapy-related myeloid neoplasms (tMN) including MDS and AML. They found that Ras pathway, RUNX1, TP53 and MLL translocations were frequent driver mutations in these patients as reminiscent with those mutation seen in adult patients with tMN. Interestingly, they found that neither of these mutations were existing as minor clones at a time of cytotoxic therapy, but seemed to occur as a consequence of the therapy. They also identified a subset of patients showing enhanced expression of MECOM/EVI1 in RNA and protein levels, of which may be activated by a translocated enhancer. Although this study is descriptive and lacking an insight into mechanism of how child hematopoietic stem cells harbored relatively less repertoires of clones at the primary

disease, compared to those in adult patients, but became more sensitive to cytotoxic therapy generating a new somatic mutation including TP53, which quickly drove the secondary malignancies, I think this study is interesting for readers working on genome and biology of myeloid malignancies, based on their finding of the difference of presence of pre-existing minor clones between child and adult patients. I wrote my questions below.

1. As the authors showed that some of patients harbored a germline mutation of TP53, but they did not mention about that of the other genes such as RUNX1 and GATA2, which are common in adult patients. It is helpful to clarify this point in the manuscript.

2. Based on previous findings, MLL fusions and the other oncogenic fusions appeared to increase expression of MECOM mRNA in patients in this study. The authors claimed that expression of MECOM mRNA was increased by translocation involving regions adjacent to ZFP36L2 at chromosome 2 and MSI2 at chromosome 17, in which the enhancer activated expression of MECOM, but this is fairly an overstatement, because the authors did not show structural association between these enhancers and the promoter of MECOM gene in cells, nor perform a functional assay to prove whether this “hijacked enhancer” contributed to increasing expression of MECOM to promote tMN. If these leukemic cells are available, the authors might want to do 3C or CRISPR to delete the enhancer region, which will strongly support their conclusion.

3. Regarding to enhanced expression of MECOM in protein, but not RNA, in blasts in a patient harboring SATB1-MECOM, is there any known mechanism to stabilize the MECOM protein, based on the altered expression of SATB1?

Response to Reviewers

We thank the reviewers for their insightful and productive comments regarding our manuscript. We have been able to sufficiently address all the comments, and the revised manuscript has clearly been strengthened.

Reviewer #1 (Remarks to the Author): Expert in tMNs and bioinformatics

Supplementary Figure 1. This distribution looks bimodal. Can you fit a two-hump distribution to it and give separate estimates of the two peaks? Does anything correlate with being in the second peak (at larger times)? I'm wondering if the gene expression pattern in the second peak supports there being more of an immune response against the clone, i.e. was this maybe the mechanism of delaying onsets of these t-MN? It would also be good to know how many background cases of t-MN were expected in the parental cohort of all children treated for 1st cancers, i.e. might these late arrivals instead really be coincidental background MN cases rather than true t-MN? The expected number needs to be calculated based on background incidence rates of 1st MN obtained from SEER and person years at risk, i.e. survival times of subjects within the much larger (~100 to 200-fold) parental cohort from which these 85 t-MN cases arose. If the number is less than 1, I'd like to know if the immune system caused the delays. If it is greater than say 10, then late arrivals are more likely to be background cases, and those with longer survival times are also more likely to be background cases (since true t-MN have worse survival), and it might nice to assign probabilities of being background cases to each person and seeing if that correlates with anything.

- i. Indeed, the age distribution of the cohort is bimodal (Dip Test $p=0.05$, Rev. Fig.1). This was true when considering all 85 cases of the cohort, and the tumor/normal cohort ($n=62$) in isolation. However, there was no significant association between age and disease-related death ($p=0.77$) or transplant-related death ($p=0.49$). There was also no difference in the age distribution amongst the pertinent molecular markers (Rev. Table 1).

Revision Figure 1

Molecular Marker	Rank-sum P value
KMT2A r	0.11
NUP98 r	0.09
del(7)	0.71
Complex karyotype	0.95
TP53 alteration	0.53
MECOM Expression (High)	0.21

Revision Table 1

- ii. Pediatric tMN is rare compared to adult tMN, therefore the background level of tMN is most likely exceedingly small/negligible. Further, calculating the expected number of tMN from pediatric cancer from our cohort/St. Jude patients is impossible, because the 85 cases included in our study do not account for every single case of tMN at St. Jude in the time period from which these cases were obtained. As referenced in the "Patient Sample Details" section of the Supplementary Methods, we only included patients with appropriate consent and suitable material available in the St. Jude Biorepository for genomic studies. We did investigate if the time from initial diagnosis to tMN was bimodal and if this time duration was associated with outcome or particular molecular features. We found that the times were not bimodal (Rev. Fig. 2, Dip test $p=0.99$) and there were no significant associations with outcome (disease-related death $p=0.72$, transplant-related death $p=0.77$) or molecular markers.
- iii. In transcriptome-wide analysis, there does not appear to be strong evidence of a broad molecular signature that associates with time interval to tMN or age at time of tMN diagnosis.

Revision Figure 2

Molecular Marker	Rank-sum P value
KMT2A r	0.41
NUP98 r	0.14
del(7)	0.6
Complex karyotype	0.72
TP53 alteration	0.12
MECOM Expression (High)	0.12

Revision Table 2

Please note that these figures have also been included as panels in Supplementary Figure 1.

Supplemental Figures 7 and 10A. The resolution here is bad. These figures should be made as pdfs (i.e. vector graphics) and read into Word as a file, not pasted in via the clipboard, as this converts them into bitmaps. Doing this will also reduce the file size.

iv. We apologize for the low image quality of these figures. These figures have been improved.

Sup Tables 2, 3, 5, and 7-11 need a freeze pane to hold the header at the top of the excel sheet.

v. Freeze panes were added for these tables.

Reviewer #2 (Remarks to the Author): Expert in paediatric and juvenile myeloid neoplasms, de novo and treatment-related

Schwartz and colleagues present a manuscript titled “The Acquisition of Molecular Drivers in Pediatric Therapy-Related Myeloid Neoplasms”. They studied a cohort of 62 patients with therapy related myeloid neoplasms (tMN) in both tumor and non-tumor material using WGS, WES, and RNAseq. 23 cases with tMN were profiled only in non-tumor specimens using WES. Common alterations included mutations in Ras pathway, *RUNX1*, and *TP53*, and *KMT2A* rearrangements (in 28/56, 60%) and high *MECOM* expression (24/56, 43%). Patients with high *MECOM* expression had either *KMT2A* rearrangements or *NUP98* fusions. Most of the mutations found in tMN were not found in primary tumors. Germline alterations with pathogenic/likely pathogenic effect were identified in 13/85 (15%) patients. While this work has technical merit and brings novel aspects to the understanding of genetics of tMN, some of the statements are misleading or inaccurate, and the manuscript is somewhat confusing.

Major comments:

1. Many findings are well known and have previously been reported, for example that complex cytogenetics is associated with poor survival, or that the tMN population is older than that of primary MDS. Also, *MECOM* locus aberrations were previously reported in tMN (PMID 22706870) in a cohort with a pediatric patient. The authors should comment on that.

vi. These points are well-received. We report these clinical characteristics and outcome findings as validation of our pediatric cohort in comparison to the previously published cohorts. We have included the Li, S., et al. citation regarding t(3;21) translocations in tMN, however, please note that this publication only includes 1 pediatric case. In our cohort, as described in the text and Supplementary Table 1, only 1 of the *MECOM* SV cases had a t(3;21) translocation, and this was associated with a *RUNX1-MECOM* fusion, confirmed by RNAseq. The other 4 *MECOM* SV cases did not have fusions, but rather SVs associated with non-coding regions that are likely enhancers, thus suggesting enhancer hijacking as the underlying mechanism of high *MECOM* expression in these cases. We believe that these cases are important to add to the current pediatric tMN literature.

2. The authors write that the prevalence of germline mutations found in this cohort (13/85, 15%) was higher than what has been previously reported for childhood cancers. This statement is misleading and “inflating the numbers”, because many of the genes shown to be mutated (Table 2) are not associated with cancers. Examples: *FANCD2* (homozygous FA, heterozygous healthy carrier) and *PMS2* (homozygous CMMRD, heterozygous Lynch syndrome). Other genes shown are dubious (*ARID2*, *EZH2*, *CREBBP*, *RPL22*, *TRIP11*). They might be associated with developmental syndromes, but are not associated with cancer. Most likely these genes were selected by the authors because they are associated with somatic changes in cancer genomes. To avoid misleading statements, the authors should report prevalence based on well-established cancer predisposition syndromes (*TP53*, *ETV6*, *NF1*, *PTPN11*), and confirm the germline status in cases with no family history/phenotype. For example patient SJ016473 with *TP53* mutation had a VAF of 7%. Was it confirmed in germline? Finally, table 6 includes 20 patients with germline mutations, but the authors write in text that 13 cases had germline mutations.

- vii. Our statement is that our finding of a 15% prevalence of germline mutations in our cohort may be more common. This statement is based on a comparison of the prevalence within our cohort and other published cohorts using an exact binomial test (see lines 93-95).
- viii. Our calculation does not include *FANCD2* and only includes likely pathogenic and pathogenic mutations (14 variants in 13 cases). These genes are the same genes used for calculations in the “Germline Mutations in Predisposition Genes in Pediatric Cancer” manuscript (see Ref. #19, Zhang et al.), which is commonly cited in the literature as a reference for germline mutations in pediatric cancers.
- ix. Indeed, heterozygous pathogenic *PMS2* variants are associated with Lynch syndrome, and Lynch syndrome is associated with an increased risk of cancer in adults (See PMID: 30376427, 29345684, 32141610).
- x. Regarding the discrepancy in the numbers (13 vs 20): Initially, Table 2 showed all P/LP variants and mosaic cases, but the calculations do not include the mosaic cases. We have removed the mosaic variants from Table 2 to decrease confusion. These variants are now separately listed in Supplementary Table 9.
- xi. SJ016473: We determined that this case was mosaic for the *TP53* variant as shown in Figure 1e. This mosaicism was confirmed in normal tissue (sorted lymphocytes) and determined to be mosaic through statistical analysis and tests (See response to Point #7 below).

3. The cohort consisted of 85 patients with tMN following hematopoietic malignancies in 60 cases, solid tumors in 23 cases, and brain tumors in 2 cases. The majority of patients (50) had ALL as primary diagnosis. This is a bias for the whole study and should be acknowledged, especially when discussing results on *KMT2A* translocations, and potentially *MECOM* expression. The authors should tone down the generalizing statements on “identifying molecular drivers in pediatric tMN” and instead clearly state that this is a single center experience with high prevalence of ALL cases and discuss how etoposide treatment could have confounded the results. It would make sense to compare the somatic profiles based on chemotherapy (etoposide yes/no).

- xii. We agree that the high percentage of cases with ALL as the primary diagnosis does potentially bias this study and has led to a high percentage of cases with *KMT2A* rearrangements. However, despite the decrease in incidence due to changes in use of topoisomerase inhibitors, tMNs driven by *KMT2A* rearrangements still occur. In Revision Figure 3 we show that overtime the frequency of *KMT2A* rearrangements decreases, likely secondary to the changes in etoposide use, but this data also shows that tMN still occurs in ALL patients of whom not all have *KMT2A* rearrangements. Revision Figure 3 shows 48 cases within our cohort who were treated on a St. Jude Total (TOT) protocols, which are traditionally named by consecutive numbers. Abbreviations: S=standard risk, H=high risk, L= low risk. Total 13B was a protocol change in Total 13 and is indicated by “TOT13B.”

Revision Figure 3

xiii. Given the frequency of *KMT2A* rearrangements in our cohort, we agree that it is important to compare the somatic profiles of those exposed to topoisomerase II inhibitors and those who were not. Revision Figure 4a shows the total mutation burden comparison between these groups. Exposure to a topoisomerase II inhibitor (etoposide or teniposide) did not significantly impact the mutation burden when considering the 4 hypermutated cases ($p=0.8208$) or when excluding them (figure not shown, $p=0.4684$). This is not surprising considering the known role of topoisomerase II inhibitors in causing double strand DNA breaks. We also tested if *KMT2Ar* were enriched in the sub-cohort being exposed to topoisomerase II inhibitors, and we did not find a significant enrichment (Fisher's exact $p = 0.1771$) (Revision Figure 4b). Interestingly, we did find that those patients never being exposed to topoisomerase II inhibitors had a significant increase in segmental copy number alterations when compared to those with exposure (Revision Figure 4c). For this comparison, whole chromosome losses or gains were not considered given that those events are not suspected to be caused by topoisomerase II inhibitors. Finally, we analyzed the WGS data via CREST from 16 cases to compare the number of structural variants between patients with topoisomerase II inhibitor exposure and those with no exposure, and we found no significant difference in the number of SVs (Revision Figure 4d). This was true even when excluding a case with chromosome 11 chromothripsis (red symbol). The Mann Whitney non-parametric test was used in panels a, c, and d.

Revision Figure 4

- xiv. Exposure to topoisomerase II inhibitors was not significantly associated with disease-related death ($p=0.91$) or transplant-related death ($p=0.50$).
- xv. Our analysis did show that MECOM expression was significantly associated with topoisomerase II inhibitor exposure ($p=0.03$) and Ras/MAPK pathway mutations ($p=0.021$). The significant association with MECOM expression is not surprising, given the known association with *KMT2Ar* (PMID: 23826732). Further, it has also previously been shown that *KMT2Ar* is associated with *Ras* mutations in precursor B-ALL (PMID: 16404744).
- xvi. We have clarified in the main text that these studies reflect the experience of a single institution (Lines 187-188).

Other comments:

4. In many patients, the number of mutations is relatively few. These numbers may not be sufficient enough to describe the significance of C>T transition among transitions and transversions (Figures 2b and 2c). Did the authors also look for the relative contribution of each transition and transversions in 16 WGS data?

xvii. **The statistical analysis used for Fig.2c was completed using the somatic tier 1 mutations in the cohort of 62 cases with tumor/normal pairs. The statistical claim is not about individual patients, but the cohort as a whole. We can very confidently say that the C>T is the most common base-pair change in the majority of patients. Further, when considering only WGS data in the 16 cases, the same trend was present.**

5. It is not clear how the authors correlate KRAS G12D variant to cisplatin signature in SJ030799 while the same variant in the patient SJ016494 was assigned to relapse MMR signature (Figure2d).

xviii. **An additional supplementary figure has been added (Supplementary Fig.8) detailing the calculations of probability that specific mutations were caused by individual mutational processes. The 2 KRAS G12D mutations are included in the examples to clearly show that they are caused by two different processes. Reference to this new supplementary figure is made in lines 123-128.**

6. The authors argue the monoallelic expression of MECOM as the supporting evidence for the cis-regulatory elements driving (enhancer high jacking) the increased levels of MECOM in ZFP36L2@-MECOM and MSI2@-MECOM. However, monoallelic expression was also observed in 4 KMT2Ar cases (Figure 3b).

We appreciate the reviewer's comments on this topic. We agree that one interesting result from this analysis was the finding of MECOM ASE in cases with a KMT2A rearrangement. While this was somewhat uncommon (4 out of 28 KMT2Ar cases), these findings may ultimately provide some mechanistic insights into the high levels of MECOM expression in KMT2Ar cases. For these cases with ASE, we speculate that the monoallelic expression of MECOM in these cases may result from epigenetic mechanisms. An additional column has been added to Supplementary Table 1 indicating with patients had MECOM ASE (column AG).

7. Supplementary methods: "Determination of mosaicism versus tumor contamination" This is confusing to a Reader. Was the mutation analysis performed in germline material or does this method refer only to analyzing hematopoietic specimens?

xix. **As discussed in the original manuscript, our germline analysis was performed using hematopoietic specimens (See Supplementary Table 1), most being sorted lymphocytes, while for some (normal tissue only patients) the specimens were from remission bone marrow. Our analysis used a previously described statistical analysis to model residual disease burden and rule out the possibility of incomplete remission and tumor-in-normal contamination (See Supplementary Ref. #16). A clarification was added to the supplementary methods section (sub-section: "Determination of mosaicism versus tumor-in-normal contamination").**

8. There are several cases with very high mutational burden. Did the authors observe chromothripsis in their study cohort?

xx. **Yes, we observed chromothripsis involving chromosome 11 in a single case within our cohort (SJ030708), and this is described in Supplementary Figure 5c. This case, SJ030708, was not a hypermutated case. In the 4 hypermutated cases, described in Supplementary Figure 2, there were other likely mechanisms (TP53 and MMR gene mutations) to explain their large mutation burden.**

9. The authors write that deletions or CN-LOH involving chromosome 7 (del(7)) were found in 22 of 62 patients (35%)". However, in Supp. Table 1, 22 of 80 patients had such lesions.

xxi. **The calculation in the text is considering only cases with tumor/normal pairs and where cytogenetic and WES/WGS copy number analysis was completed. The additional 23 normal only cases were not included in this calculation given that the same type of data was not available.**

10. *Supp. Table 2 (somatic) and Table 9 (germline): it is not clear what material was used as germline control.*

xxii. **A description of the germline source is described in the supplementary methods section (sub-section: “Patient sample details”). Additionally, column C (“Normal gDNA Source”) has been added to Supplementary Table 1.**

11. *Supp. Figure 4b: based on tumor VAF the authors suggest that TP53 are early and PPM1D mutations are late, but no additional data is provided to support this statement. I suggest verifying this using single cell studies, or at least longitudinal studies.*

xxiii. **We apologize for any confusion. As shown in Figure 4b, the TP53 variants typically have a VAF that overlaps or is higher (due to loss of heterozygosity) than the mean VAF for the other somatic mutations in each case. This very finding implies that the variant is present in most of the cells in the tumor. While using bulk VAFs to impute the timing of variant acquisition is somewhat common, we agree with the reviewer that single cell sequencing is the ideal way to illustrate this finding. However, this distinction isn’t a major tenet of this study and we have removed this statement to avoid any unnecessary confusion.**

12. *Supp. Figure 8: the bottom gel is not clear. What primers were used, what length was expected? Were the products sequenced? I suggest repeating this experiment and sequencing the products to confirm the fusion sequence.*

xxiv. **Supplementary Figure 8b has been added showing primers used for fusion validation. The products were not sequenced. We are unable to repeat the experiment given that the starting amount of patient material was minimal and there is none remaining. Please also note that the majority of these fusions had orthogonal validation in the form of cytogenetics or WGS.**

Minor comments:

1. *Line 93, pathologic or likely pathologic can be written as pathogenic or likely pathogenic (these are most widely used terms)*

xxv. **This change has been made.**

2. *Figures 3d,4c, and 4d are not referred to in the manuscript.*

xxvi. **These figures are now referred to in the manuscript.**

3. *Supplementary figure 7- x-axis is not readable.*

xxvii. **Figure has been replaced with an .svg file, improving resolution.**

4. *In the supplementary figures 3, 4 and 9, the term secondary protein structure is used. This has to be corrected as protein domain architecture.*

xxviii. **This correction has been made.**

5. *A number of typos and incomplete figure legends. For example what is “*” and “Y” in Supp Fig 3d*

xxix. **Figure legends have been reviewed. For Supplementary Fig. 3d those symbols were originally labeled in the figure—“*= mosaicism, Y= germline.”**

6. *Many unclear abbreviations in Supp. Table 17*

xxx. **An abbreviation legend has been placed at the bottom left of Supplementary Table 17.**

7. *Fig. 3e: SJ030708 panel is not visible*

xxxi. **We apologize for the confusion. SJ030708 was included as a negative control to show the low level of background staining in cases without elevated MECOM expression. This has been made clearer in the legend for Figure 3.**

8. *What does the finding of C>T transitions mean? This is a known mutational signature of UV light / epithelial cancers. The authors should discuss this.*

xxxii. **See response to point #4 (xi). It is also important to note that we did not only analyze C>T transitions but rather we describe a comprehensive mutational signature analysis of 96 trinucleotide profiles across the samples within the tumor/normal cohort (See Supplementary Methods section “Mutational signature analysis.”**

In this study, the authors performed comprehensive analyses of genomic mutations in pediatric therapy-related myeloid neoplasms (tMN) including MDS and AML. They found that Ras pathway, RUNX1, TP53 and MLL translocations were frequent driver mutations in these patients as reminiscent with those mutation seen in adult patients with tMN. Interestingly, they found that neither of these mutations were existing as minor clones at a time of cytotoxic therapy, but seemed to occur as a consequence of the therapy. They also identified a subset of patients showing enhanced expression of MECOM/EVI1 in RNA and protein levels, of which may be activated by a translocated enhancer. Although this study is descriptive and lacking an insight into mechanism of how child hematopoietic stem cells harbored relatively less repertoires of clones at the primary disease, compared to those in adult patients, but became more sensitive to cytotoxic therapy generating a new somatic mutation including TP53, which quickly drove the secondary malignancies, I think this study is interesting for readers working on genome and biology of myeloid malignancies, based on their finding of the difference of presence of pre-existing minor clones between child and adult patients. I wrote my questions below.

1. As the authors showed that some of patients harbored a germline mutation of TP53, but they did not mention about that of the other genes such as RUNX1 and GATA2, which are common in adult patients. It is helpful to clarify this point in the manuscript.

xxxiii. **We mentioned in our original submission (line 104) that germline GATA2 mutations were not present in our cohort. We have added RUNX1 (See line 106-107 in the revised manuscript).**

2. Based on previous findings, MLL fusions and the other oncogenic fusions appeared to increase expression of MECOM mRNA in patients in this study. The authors claimed that expression of MECOM mRNA was increased by translocation involving regions adjacent to ZFP36L2 at chromosome 2 and MSI2 at chromosome 17, in which the enhancer activated expression of MECOM, but this is fairly an overstatement, because the authors did not show structural association between these enhancers and the promoter of MECOM gene in cells, nor perform a functional assay to prove whether this “hijacked enhancer” contributed to increasing expression of MECOM to promote tMN. If these leukemic cells are available, the authors might want to do 3C or CRISPR to delete the enhancer region, which will strongly support their conclusion.

xxxiv. **The integration of the WGS and RNASeq data clearly shows that there is a structural variant adjacent to MECOM, that this structural variant is specific to cases with the highest MECOM expression, and that there is corresponding high and allele specific expression. These findings are consistent with other studies describing these types of events. For instance, a recent publication from Ottema et al. (PMID: 32219447). It would be an illustrative experiment to silence the enhancer on chromosome 2, via CRISPRi, in patient samples, and subsequently evaluate MECOM expression. Unfortunately, conducting these studies in patient samples, which are limiting, is currently not feasible. In addition, we have not been successful in generating PDX lines from these tMN patient samples and we are not aware of any cell line that harbors this chr2-chr3 event. However, we have completed additional *in silico* analyses of published CHIP-seq, ATAC-Seq, and CTCF CHIP-seq in human CD34 cells (PMID: 32386543, 32386543, & 32386543) which support our claims that the genomic locations involved in those MECOM SVs in our cohort are within/adjacent to enhancers in hematopoietic progenitors and are likely driving high expression of MECOM. We show these analyses in a new supplementary figure (Supplementary Fig. 13).**

3. Regarding to enhanced expression of MECOM in protein, but not RNA, in blasts in a patient harboring SATB1-MECOM, is there any known mechanism to stabilize the MECOM protein, based on the altered expression of SATB1?

xxxv. **MECOM (EVI1) is a C2H2 zinc finger containing protein that is part of a large complex of proteins that regulate transcription. While its activity has been shown to be regulated by protein phosphorylation (PMID 23858473), there is very little known about the stability of the MECOM (or EVI1) protein. The mechanism by which this case with a clear MECOM SV and elevated protein levels, yet relatively low MECOM RNA levels, is currently unknown.**

REVIEWERS' COMMENTS

Reviewer #1 (Remarks to the Author):

Perhaps a crude upper bound could be obtained based on numbers of 1st cancer cases seen on average per year at St. Jude and their average life expectancies, making a simplifying assumptions of a uniform distribution of ages at diagnosis ... I don't think it is completely impossible to say something about the order of magnitude of an upper limit of the expected numbers of background MN cases falsely classified as t-MN.

Reviewer #2 (Remarks to the Author):

I appreciate the authors answers to my questions and have no further comments.

Reviewer #3 (Remarks to the Author):

The authors have clearly responded to my comments in this revised manuscript.

Response to Reviewers

We again thank the reviewers for their review of our revision and the additional productive comments regarding our manuscript. We have addressed the follow up comment from Reviewer #1.

Reviewer #1 (Remarks to the Author):

Perhaps a crude upper bound could be obtained based on numbers of 1st cancer cases seen on average per year at St. Jude and their average life expectancies, making a simplifying assumptions of a uniform distribution of ages at diagnosis. I don't think it is completely impossible to say something about the order of magnitude of an upper limit of the expected numbers of background MN cases falsely classified as t-MN.

- i. The age distribution of patients at the time of their primary oncologic diagnosis is bimodal in this study (Rev.Fig.1a). Similarly, the age distribution at the time of tMN is bimodal (Rev.Fig.1b). The upper age limit within a therapy related myeloid neoplasm study will be dependent on the age at which the primary oncologic diagnosis occurs. The published studies to date report the latency as defined by the time to onset of tMN in order to establish a relationship to the prior cytotoxic chemotherapy and/or radiation. Our median age at diagnosis of tMN is 13.6 years with a median latency of 4.0 years (Rev.Fig.1c). This is consistent with previously published reports including the MD Anderson experience with pediatric tMN that found a median age at diagnosis of 14 years and a median latency of 4.1 years¹. Similar data has been published in the adult setting with a medium latency of 3.9 years². Eight patients in our study had a latency greater than 10 years. Six of these patients had significant exposure to etoposide, alkylating agents, radiation, or all three. Among these 8 patients there are no survivors. In contrast, outcomes for patients with *de novo* AML less than 21 years of age is approximately 60% with current standard of care. We would expect at least one or two of these eight patients to have survived if, in fact, these are *de novo* malignancies. Given the significant exposure, poor outcome, and prior history of malignancy, we feel it is unlikely that these patients are *de novo* myeloid neoplasms. Please note that in the aforementioned adult study, the latency range was 0.5-22 years. Therefore, longer latencies have been reported.

Revision Figure 1

- ii. Further, we computed an estimated age of myeloid neoplasm for each subject based on the age of first cancer and the published age-specific incidence rates for AML in the SEER database³ (https://seer.cancer.gov/explorer/application.html?site=96&data_type=1&graph_type=3&compareBy=sex&chk_sex_3=3&chk_sex_2=2&race=1&hdn_rate_type=1&advopt_precision=1) for ages 0-25 years. The results are shown in Revision Figure 2. Fifty-five of 62 (88%; 95% CI: 78.1%-95.3%) patients developed a myeloid neoplasm before the estimated median age from SEER ($p = 2.43 \times 10^{-10}$, sign-test). This analysis supports the interpretation that most of the cohort is correctly considered to have tMN rather than a secondary “background” AML. This analysis has been included as Supplementary Figure 1e.

Revision Figure 2

Rev.Fig.2. Each row of three points represents the age of first cancer diagnosis (black dot), the actual age of MN diagnosis (red dot), and the estimated median age of MN diagnosis based on age of first cancer diagnosis from SEER data (gray dot). Patients are sorted by age of first diagnosis. Fifty-five of 62 patients developed a myeloid neoplasm earlier than estimated from SEER incidence data.

REFERENCES

1. Aguilera, D.G. *et al.* Pediatric therapy-related myelodysplastic syndrome/acute myeloid leukemia: the MD Anderson Cancer Center experience. *J Pediatr Hematol Oncol* **31**, 803-11 (2009).
2. Schoch, C., Kern, W., Schnittger, S., Hiddemann, W. & Haferlach, T. Karyotype is an independent prognostic parameter in therapy-related acute myeloid leukemia (t-AML): an analysis of 93 patients with t-AML in comparison to 1091 patients with de novo AML. *Leukemia* **18**, 120-5 (2004).
3. SEER*Explorer: An interactive website for SEER cancer statistics. (Surveillance Research Program, National Cancer Institute, 2020).